METHODS

# A deep learning approach for time-consistent cell cycle phase prediction from microscopy data

**Thomas Bonte**[1,2,3], **Oriane Pourcelot**[4], **Adham Safieddine**[5,6], **Floric Slimani**[4], **Florian Mueller**[7], **Dominique Weil**[5,6], **Edouard Bertrand**[4], **Thomas Walter**[1,2,3]*

**1** Center for Computational Biology, Mines Paris PSL, Paris, France, **2** U1331 - Computational Oncology, Institut Curie, Paris, France, **3** U1331 - Computational Oncology, INSERM, Paris, France, **4** IGH, CNRS and Montpellier University, Montpellier, France, **5** Development, Adaptation and Ageing (Dev2A), Sorbonne Université, CNRS, INSERM, Paris, France, **6** Institut de Biologie Paris-Seine (IBPS), Sorbonne Université, CNRS, INSERM, Paris, France, **7** Institut Pasteur, Université Paris Cité, Centre de Ressources et Recherches Technologiques, UTechS-PBI, C2RT, Paris, France

* thomas.walter@minesparis.psl.eu

## Abstract

The cell cycle is a series of regulated stages during which a cell grows, replicates its DNA, and divides. It consists of four phases – two growth phases (G1 and G2), a replication phase (S), and a division phase (M) – each characterized by distinct transcriptional programs and impacting most other cellular processes. In imaging assays, the cell cycle phase can be identified using specific cell-cycle markers. However, the use of dedicated cell-cycle markers can be impractical or even prohibitive, as they occupy fluorescent channels that may be needed for other reporters. To address this limitation we propose a method to infer the cell cycle phase from a widely used fluorescent reporter: SiR-DNA, thereby bypassing the need for phase-specific markers while leveraging information already present in common experimental setups. Our method is based on a Variational Auto-Encoder (VAE), enhanced with two auxiliary tasks: predicting the average intensity of phase-specific markers and enforcing temporal consistency through latent space regularization. The reconstruction task ensures that the latent space captures cell cycle–relevant features, while the temporal constraint promotes biological plausibility. The resulting model, CC-VAE, classifies cell cycle phases with high accuracy from widely used DNA markers and can thus be applied to high-content screening datasets not specifically designed for cell cycle analysis. CC-VAE is freely available, along with a new, publicly released dataset comprising over 600,000 labeled HeLa Kyoto nuclear images to support further development and benchmarking in the community.

**Data availability statement:** The datasets generated and analysed during the current study are available in the BioImage Archive repository, https://www.ebi.ac.uk/biostudies/bioimages/studies/S-BIAD1659. The Python code is available via GitHub, https://github.com/15bonte/cell_cycle_classification. The tuned models are available via HuggingFace, https://huggingface.co/thomas-bonte/cell_cycle_classification.

**Funding:** This work has been supported by the French government under management of Agence Nationale de la Recherche (ANR) as part of the "Investissements d'avenir" program, reference ANR-19-P3IA-0001 (PRAIRIE 3IA Institute; TW), the France 2030 program PRAIRIE-PSAI with reference number ANR-23-IACL-0008 (TW), and the ANR project TRANSFACT, reference ANR-19-CE12-0007 (EB, FM, TW). Furthermore, we also acknowledge support by France-BioImaging, reference ANR-10-INBS-04 (EB). Furthermore, this work was supported by a government grant managed by the Agence Nationale de la Recherche under the France 2030 program, with the reference numbers ANR-24-EXCI-0001 (EB, FM, TW), ANR-24-EXCI-0002 (EB, FM, TW), ANR-24-EXCI-0003 (EB, FM, TW), ANR-24-EXCI-0004 (EB, FM, TW), ANR-24-EXCI-0005 (EB, FM, TW). The funders had no role in study design, data collection and analysis, decision to publish, or preparation of the manuscript. The authors received no specific funding for this work.

**Competing interests:** The authors have declared that no competing interests exist.

## Author summary

The cell cycle is a fundamental biological process involving distinct stages of growth, DNA replication, and division. Understanding which stage a cell is in can help researchers study many biological phenomena, including how cells respond to treatments. Traditionally, identifying these stages requires adding special fluorescent markers to cells that report on cell cycle progression. However, such markers are not always practical and can limit the experimental readout. In our study, we developed a method to determine a cell's cycle phase using a common DNA stain that is already employed in many imaging experiments. Our approach maps cell images into a latent space that is both informative of the cell cycle phase and temporally consistent: images of the same cell taken at consecutive time points are mapped to nearby points in this space. From this latent space, we can accurately recognize cell cycle stages without the need for additional markers. Alongside this method, which we make freely available, we also created and released a large, labeled image dataset to support further research. This tool can help scientists extract more value from their imaging data and gain insights from existing experiments.

## 1 Introduction

The cell cycle is a tightly regulated series of stages resulting in cell growth, DNA replication, organelle duplication, and the partitioning of cellular components through cell division. The cell cycle consists of four distinct phases: G1 (gap 1), S (synthesis), G2 (gap 2), and M (mitosis), each characterized by specific gene expression programs. Throughout these phases, cells dynamically regulate gene expression, protein synthesis, and intracellular organization to ensure proper division and function. Many aspects of a cell depend directly or indirectly on its cell cycle phase, including gene expression [1], transcriptional activity and histone modification [2], protein interaction states [3] and drug sensitivity [4,5]. Therefore, interpreting observations, such as localization patterns of RNAs and proteins or cellular phenotypes in the context of the cell cycle phase has the potential to reveal important functional dependencies that might otherwise remain hidden.

Cellular imaging has been widely used to investigate the cell cycle, often relying on fluorescent markers such as FUCCI, whose expression levels are correlated with the cell cycle phase [6–9]. However, such markers are invasive and may introduce experimental biases [10,11]. Moreover, they occupy fluorescent channels that might be essential for other readouts when the primary focus is not the cell cycle itself. For instance, in studies of RNA [12] or protein localization [13], it may be important to examine how localization patterns depend on the cell cycle. In these cases, reserving two fluorescent channels for cell cycle markers reduces the ability to simultaneously capture other relevant signals. This limitation is particularly problematic when cell cycle information is needed as a complementary feature in assays focused on other biological processes.

Deep learning, and representation learning in particular, has demonstrated remarkable efficiency in extracting subtle cellular morphological features to significantly advance our understanding of biological processes [14–21]. Representation learning is a self-supervised learning approach designed to capture rich representations directly from data without prior assumptions or need for annotations. It extracts meaningful features from cellular phenotypes, which can then be used for various downstream tasks both at single-cell and subcellular resolutions: clustering [16,18,21], cell state classification [15,17], cell cycle phase classification [19,20], subcellular protein localization [18,19].

Several recent studies have proposed deep and/or representation learning strategies for performing cell cycle phase classification without dedicated cycle markers. Deep Convolutional Neural Networks (CNN) have been proposed for cell cycle phase classification, learning complex patterns from labeled microscopy images in a fully supervised manner [22–25], operating either on label-free or fluorescence microscopy. Alternatively, meaningful hand-crafted features can be extracted from microscopy data for classification using Support Vector Machine (SVM) [25,26]. Another approach is self-supervised learning which reduces the need for large training datasets while enabling more generalizable and robust feature extraction [19,20]. The extracted features can then be used to train a (small) classification network to predict the cell cycle phases. Finally, [27] uses a U-Net model to directly predict a cell cycle mask.

However, the majority of these approaches simplify the classification task by merging distinct cell cycle phases. These phases are either grouped together into a single class, referred to as *interphase* [19,22] or split into two broader categories for binary classification: G1/S vs G2/M [24] or G1 vs S/G2/M [25,26]. Such an approach overlooks the finer distinctions between the phases, which are crucial for accurately understanding cell cycle dynamics. Among the works that do not merge G1, S and G2/M, [23] proposes a method to reconstruct a cyclic latent space based on the classification of four manually defined virtual phases. However, this approach struggles to accurately distinguish between the G1, S, and G2/M phases, limiting its effectiveness for precise cell cycle phase classification. [27] presents a method for cell cycle phase classification using Spatial Light Interference Microscopy (SLIM, [28]) images. Although the approach achieves decent classification accuracies, SLIM is not widely used, which limits the general applicability of the method. Finally, [20] employs a VQ-VAE combined with Dynamic Time Warping (DTW) to classify G1, S, G2, and M phases separately. While the approach achieves a classification accuracy similar to [27], it requires live-cell imaging data, as DTW is used in conjunction with deep learning. Consequently, it does not provide a solution for single-image classification tasks.

Each of these approaches thus comes with specific limitations, as they either rely on specific experimental setups, such as live-cell imaging [20] or advanced imaging technologies [27], or because they simplify the problem by merging phases (e.g., G1/S vs. G2/M) [19,22,24]. At the same time, insights from live-cell imaging studies demonstrate that temporal coherence—the smooth and consistent evolution of cellular states over time—provides an essential, complementary perspective beyond simple phase classification. Capturing this continuity is key to faithfully modeling the cell cycle as a gradual process rather than a sequence of discrete categories.

Motivated by these observations, we propose to leverage representation learning to infer the cell cycle phase from a single fluorescence image of a DNA marker, without requiring live-cell tracking. Commonly used DNA markers such as DAPI, Hoechst, or SiR-DNA [29] contain rich structural information that reflects nuclear morphology and chromatin organization across the cell cycle. During the S phase, for instance, DNA replication doubles the chromatin content, leading to characteristic changes in nuclear size and fluorescence intensity [30–32]. These variations, together with chromatin texture patterns, encode cell-cycle-dependent features that conventional analyses have struggled to fully exploit.

Thus, our goal is to provide the community with a robust, simple, and open-source method for cell cycle phase classification, along with a large publicly available nucleus image dataset to support further research. Specifically, we introduce Cell Cycle Variational Auto-Encoder (CC-VAE), a new self-supervised strategy based on a Variational Auto-Encoder (VAE, [33]) to learn a meaningful and biologically consistent latent representation of nuclei SiR-DNA images and accurately perform cell cycle phase classification.

In the following, we present our new image dataset and provide an overview of the CC-VAE methodology. We then explore the latent space of the nucleus representations learned by CC-VAE and assess its ability to accurately classify

cell cycle phases. This performance is further evaluated through benchmarking against alternative models and ablation studies.

## 2 Results

### 2.1 Dataset

We introduce a new image dataset, which comprises 982,332 images of HeLa Kyoto nuclei including 636,304 labeled into one of the three classes G1, S or G2/M. Images were acquired with a 63× objective microscope. Each image contains five focal planes and three channels: a DNA marker (SiR-DNA) and two PIP-FUCCI markers derived from the PIP-FUCCI system [9]: mCherry-Gem$_{1-110}$ and PIP-mVenus. These markers allow for the ground truth inference of the cell cycle phase for each individual cell.

We performed time-lapse microscopy via live-cell fluorescence imaging of HeLa Kyoto cells labeled with SiR-DNA and the two PIP-FUCCI markers. Images were captured every 20 minutes over a period of 40 hours. While this temporal resolution is crucial for confidently assessing the cell cycle phase for each cell, it is important to note that our classification method is designed to process a single image as input, here the DNA-labeled cells. Cell cycle annotation was based on the temporal evolution of the PIP-FUCCI intensities across their area. By doing this, the cell cycle phase was determined for 636,304 nuclei. 346,028 nuclei remained unlabeled due to the inability to reliably determine cell cycle phase boundaries based on their PIP-FUCCI intensity changes, as the tracks were either too short, displayed only one of the three phases, or were truncated at the end of the video.

Fig 1 displays some example nuclei from our annotated dataset. G1 nuclei express only PIP-mVenus fluorescence, S nuclei express only mCherry-Gem$_{1-110}$ fluorescence, and G2 nuclei express both PIP-FUCCI signals [9].

### 2.2 Overview of CC-VAE

Our method, CC-VAE, is based on a customized $\beta$-Variational Auto-Encoder ($\beta$-VAE, [34]) architecture to learn an unsupervised latent representation of nucleus images (see Fig 2A).

Vanilla VAEs [33] compress an original image $\mathbf{x}$ into its latent representation $\mathbf{z} \in \mathbb{R}^d$ through the encoder $q_\phi$. The decoder $p_\theta$ then uses this latent representation as input to reconstruct the initial image, $\hat{\mathbf{x}}$. By constraining the latent distribution $q_\phi(\mathbf{z}|\mathbf{x})$ to follow a standard Gaussian $\mathcal{N}(\mathbf{0}, \mathbf{I_d})$, VAEs learn a robust and powerful latent representation. Both the encoder and decoder are parameterized by deep neural networks. This allows VAEs to be trained by minimizing the sum of a reconstruction loss, $\mathcal{L}_{\text{REC}} = -\mathbb{E}_{\mathbf{z} \sim q_\phi}[\log p_\theta(\mathbf{x}|\mathbf{z})] = \frac{1}{2}\text{MSE}(\mathbf{x}, \hat{\mathbf{x}})$, and a regularization loss, $\mathcal{L}_{\text{REG}} = \text{KL}(q_\phi(\mathbf{z}|\mathbf{x})\|\mathcal{N}(\mathbf{0}, \mathbf{I_d}))$ – where MSE and KL refer to the Mean-Square Error and the Kullback-Leibler divergence, respectively. As [34] introduced a parameter $\beta$ to balance between reconstruction accuracy and latent space regularization, $\beta$-VAE are trained by minimizing the loss function:

$$
\begin{aligned}
\mathcal{L}_{\beta\text{-VAE}} &= \mathcal{L}_{\text{REC}} + \beta \cdot \mathcal{L}_{\text{REG}} \\
&= \frac{1}{2}\text{MSE}(\mathbf{x}, \hat{\mathbf{x}}) + \beta \cdot \text{KL}(q_\phi(\mathbf{z}|\mathbf{x})\|\mathcal{N}(\mathbf{0}, \mathbf{I_d}))
\end{aligned}
\tag{1}
$$

Building on this approach, we aimed to adapt it for the task of learning time-consistent, cell cycle-aware latent representations of cell nuclei.

First, we use the PIP-FUCCI channels during the self-supervised training. Inspired by in-silico labeling strategies described in [35,36], we utilize our latent representation to predict the average intensity of both PIP-FUCCI signals across the nucleus area. Since the total fluorescent intensities are closely associated with cell cycle phases, this additional task helps structure the latent space to better support downstream cell cycle phase classification.

Second, we exploit the time dependency of our images to enforce time consistency within the latent space. Contrastive learning [37] aims at learning rich image representations by generating multiple views of the same input image using

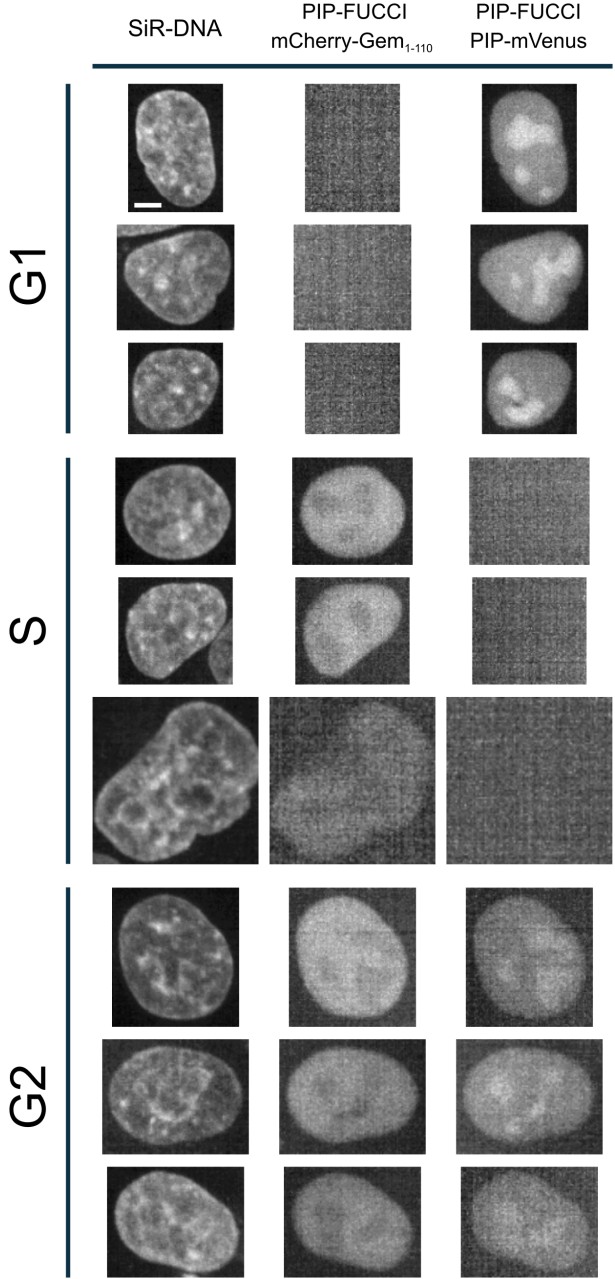

**Fig 1. Example nuclei from our annotated dataset.** G1 nuclei express only PIP-mVenus fluorescence, S nuclei express only mCherry-Gem$_{1\text{-}110}$ fluorescence, and G2 nuclei express both signals. Scale bar: 5μm.

semantic-preserving data augmentations, and train models to group those views together in representation space. If two views come from the same input image (defining a positive pair), their representations should be close; otherwise (negative pair), their representations should be dissimilar and thus more distant. Here, we assume that the nuclei do not drastically change from one time frame to another. We therefore expect that the representations of one nucleus at two consecutive time frames should be relatively close in the latent space. This one-frame shift can therefore be seen as a

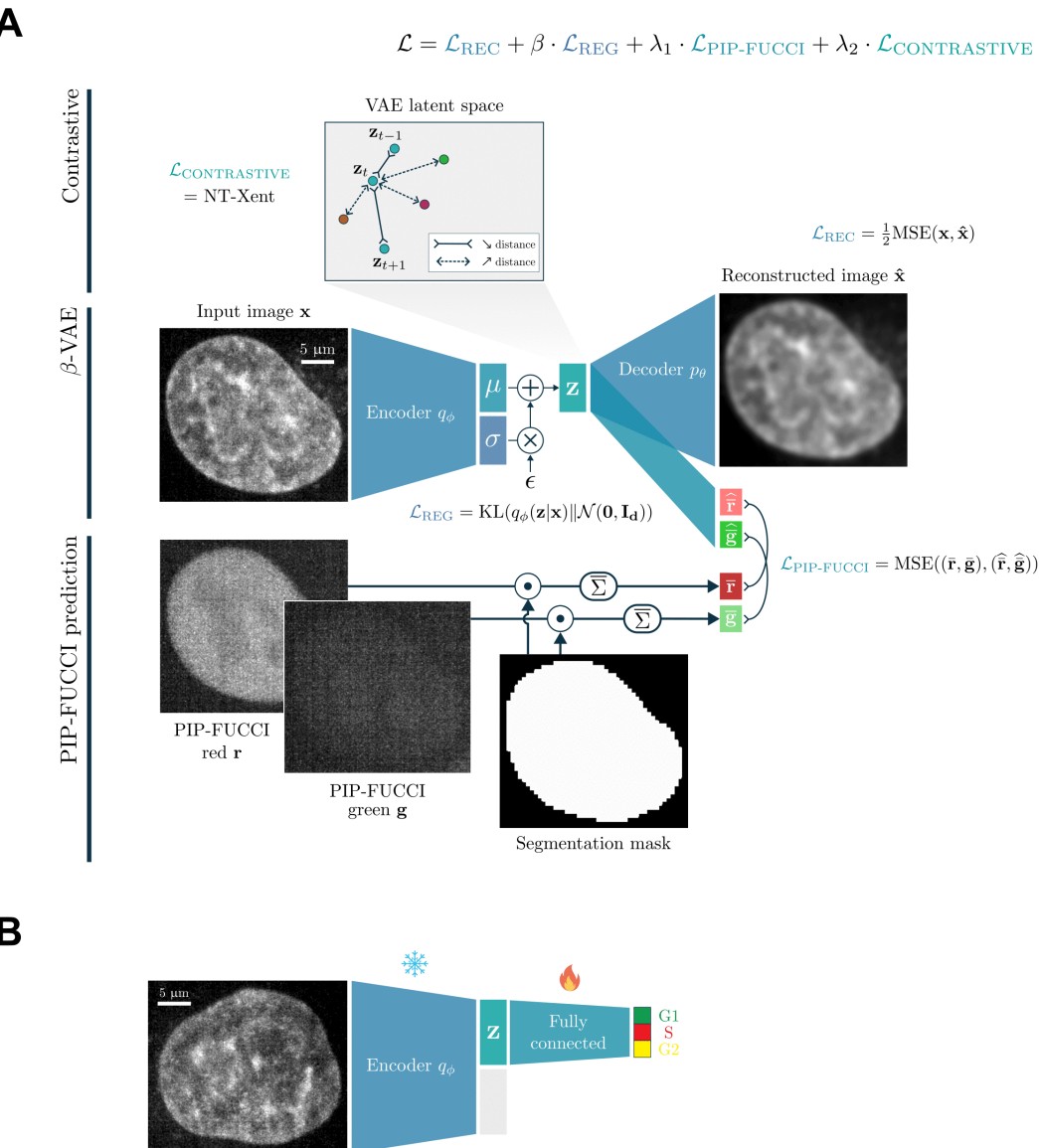

**Fig 2. Architecture of CC-VAE.** (A) Building on the $\beta$-VAE architecture, we enhance the model by predicting the average intensity of both PIP-FUCCI signals from the latent representation and ensuring temporal consistency through a contrastive loss applied in the latent space. (B) For classification, two fully connected layers are added on top of the frozen CC-VAE encoder to perform supervised training.

semantic-preserving image augmentation. Consecutive frames of the same nucleus would be treated as positive pairs, while all other nuclei within the training batch would be treated as negative samples. If the frame is the first or last in its sequence, only one positive pair is considered for it. To encourage such positive pairs to move towards each other and negative pairs to move away from each other, we incorporate the Normalized Temperature-Scaled Cross-Entropy (NT-Xent) loss, as proposed in [38], into our training process. For a positive pair of examples $(i,j)$, this loss function writes:

$$l_{i,j} = -\log \frac{\exp(\mathrm{sim}(\mathbf{z}_i, \mathbf{z}_j)/\tau)}{\sum_k \mathbb{1}_{(k,i)\notin P} \exp(\mathrm{sim}(\mathbf{z}_i, \mathbf{z}_k)/\tau)} \tag{2}$$

where sim refers to the cosine similarity function, $\tau$ is the temperature parameter, and $P$ is the set of positive pairs within the corresponding training batch. $\mathbb{1}_{(k,i)\notin P}$ is the indicator function evaluating 1 iff $(k,i)\notin P$. The contrastive loss is computed across all positive pairs in the batch: $\frac{1}{2|P|}\sum_{(i,j)\in P}(l_{i,j}+l_{j,i})$.

Our final loss function is thus:

$$
\begin{aligned}
\mathcal{L}_{\text{CC-VAE}} =&\mathcal{L}_{\beta\text{-VAE}} + \lambda_1 \cdot \mathcal{L}_{\text{PIP-FUCCI}} + \lambda_2 \cdot \mathcal{L}_{\text{CONTRASTIVE}} \\
=&\frac{1}{2}\text{MSE}(\mathbf{x}, \hat{\mathbf{x}}) + \beta \cdot \text{KL}(q_\phi(\mathbf{z}|\mathbf{x})\|\mathcal{N}(\mathbf{0},\mathbf{I_d})) \\
&+ \lambda_1 \cdot \text{MSE}((\bar{\mathbf{r}},\bar{\mathbf{g}}),(\hat{\bar{\mathbf{r}}},\hat{\bar{\mathbf{g}}})) \\
&+ \lambda_2 \cdot \text{NT-Xent}
\end{aligned}
\tag{3}
$$

where $(\bar{\mathbf{r}},\bar{\mathbf{g}})$ and $(\hat{\bar{\mathbf{r}}},\hat{\bar{\mathbf{g}}})$ are the true and predicted average PIP-FUCCI mCherry-Gem$_{1\text{-}110}$ (red) and PIP-mVenus (green) intensities, respectively.

Following this self-supervised training, we train two fully connected layers on top of the frozen CC-VAE encoder features to predict the cell cycle phase for a given nucleus (see Fig 2B).

## 2.3 CC-VAE learns rich nucleus representations

CC-VAE allows us to map each nucleus image to a representation in a latent space. Fig 3 shows the UMAP representation of the learned embeddings from a subset of the test set, specifically 276 cell tracks that comprise all three cell cycle phases G1, S and G2/M. Each dot represents a nucleus. In particular, the nuclei presented in Fig 1 are localized in the latent space. Even though the cell cycle information was not used at all during self-supervised training, CC-VAE effectively disentangles the latent space, such that each phase corresponds to coherent regions.

Fig 4 displays the same UMAP representation, with nuclei colored according to different rules. In Fig 4B, nuclei are colored depending on the time elapsed since mitosis. CC-VAE successfully captures this temporal progression: early-cycle nuclei are positioned at the edges of the latent space, while late-cycle nuclei cluster toward the center. This spatial organization highlights the model's ability to encode temporal information related to cell cycle progression in its learned representations.

Fig 4C and 4D present the latent space with nuclei colored by their area and total SiR-DNA intensity normalized across all nuclei, respectively. In Fig 4C, larger nuclei locate at the center and top of the latent space, which corresponds to the G2/M phase. This placement aligns with the expected biological progression, as nuclei increase in size throughout the cell cycle, reaching their maximum dimensions during the G2/M phase [30–32]. In Fig 4D, nuclei with higher SiR-DNA intensity are located on the top-right side of the latent space, corresponding to the S and G2/M phase.

Additionally, our model identifies specific nuclei that would have been misclassified if only area and SiR-DNA intensity were considered. The small cluster of nuclei marked by the blue arrow in Fig 4A and 4C represents large S-phase nuclei, which could be mistaken for G2/M nuclei due to their size. Similarly, nuclei indicated by the orange arrow in Fig 4A, 4B, and 4C correspond to small early S-phase nuclei, which could be incorrectly classified as G1 nuclei.

As our dataset is derived from live-cell imaging, it is possible to compute the latent representation of all nuclei within a specific track, allowing to trace the cell trajectory in the latent space. This representation visualizes the nucleus evolution throughout the cell cycle. Fig 5 displays six distinct nucleus trajectories transitioning from G1 to G2/M. This visualization highlights the temporal consistency achieved by our approach, where nuclei that are phenotypically close (such as the same nucleus in subsequent time frames) are also close in latent space.

   

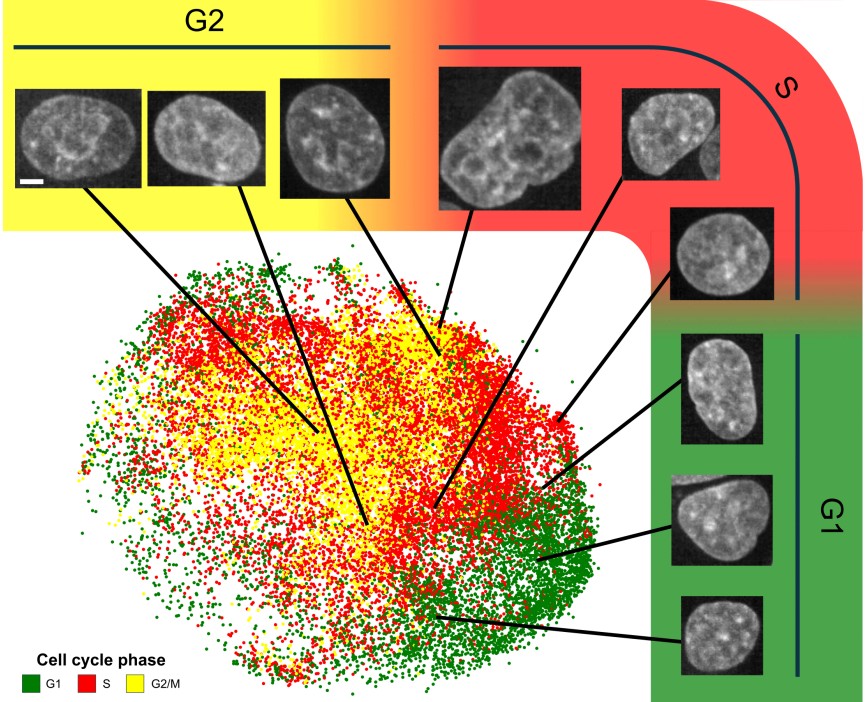

**Fig 3**. **UMAP representation of the nucleus representations learned by CC-VAE.** Each dot corresponds to a nucleus, colored by cell cycle phase. Scale bar: 5µm.

## 2.4 CC-VAE classifies cell cycle phases with high accuracy

Given the structure of the latent space, the learned latent representations can be used to train a robust cell cycle phase classifier. Fig 6A shows the averaged confusion matrix from our 10-fold cross-validation dataset (see Methods). We observe good performance with an average recall of 82% across classes (86% for G1, 77% for S and 83% for G2/M) and average precision of 82% (88% for G1, 80% for S and 78% for G2/M).

Fig 6B presents the nucleus area as a function of the predicted cell cycle phase on unseen test data. As expected, nucleus size increases progressively throughout the cell cycle, consistent with previous observations reported in the literature [30–32]. In particular, Fig 6B reproduces the trend shown in Fig 4C of [31], illustrating the increase in nuclear size from G1 to S phase and from S to G2/M.

As expected, misclassifications between G1 and G2/M nuclei are rare (only 2.5% of all errors) due to their distinct features. Although the S phase remains the most challenging to classify due to overlapping boundaries with both G1 (42.3% of errors are G1 vs S confusions) and G2/M (55.2% of errors were G2/M vs S confusions), our approach demonstrates reliable performance with 77% recall and 80% precision for S as the most difficult class.

Fig 7A shows the UMAP representation of nuclei from the same subset of the test set as in Fig 3, with both ground truth and predicted cell cycle phases. Specifically, Fig 7B focuses on the blue and orange regions highlighted in Fig 4A, demonstrating overall strong classification accuracy in these phase-transition areas. However, our model is not entirely free from misclassifications. Fig 7C, 7D focus on the classification errors within these two regions, along with representative examples of misclassified nuclei. As expected, these include large S-phase nuclei misclassified as G2/M and small early S-phase nuclei incorrectly labeled as G1. Because their nuclear areas slightly deviate from the typical phase distributions, CC-VAE had difficulty classifying them with full confidence. Similarly, some small G2/M nuclei and large G1 nuclei

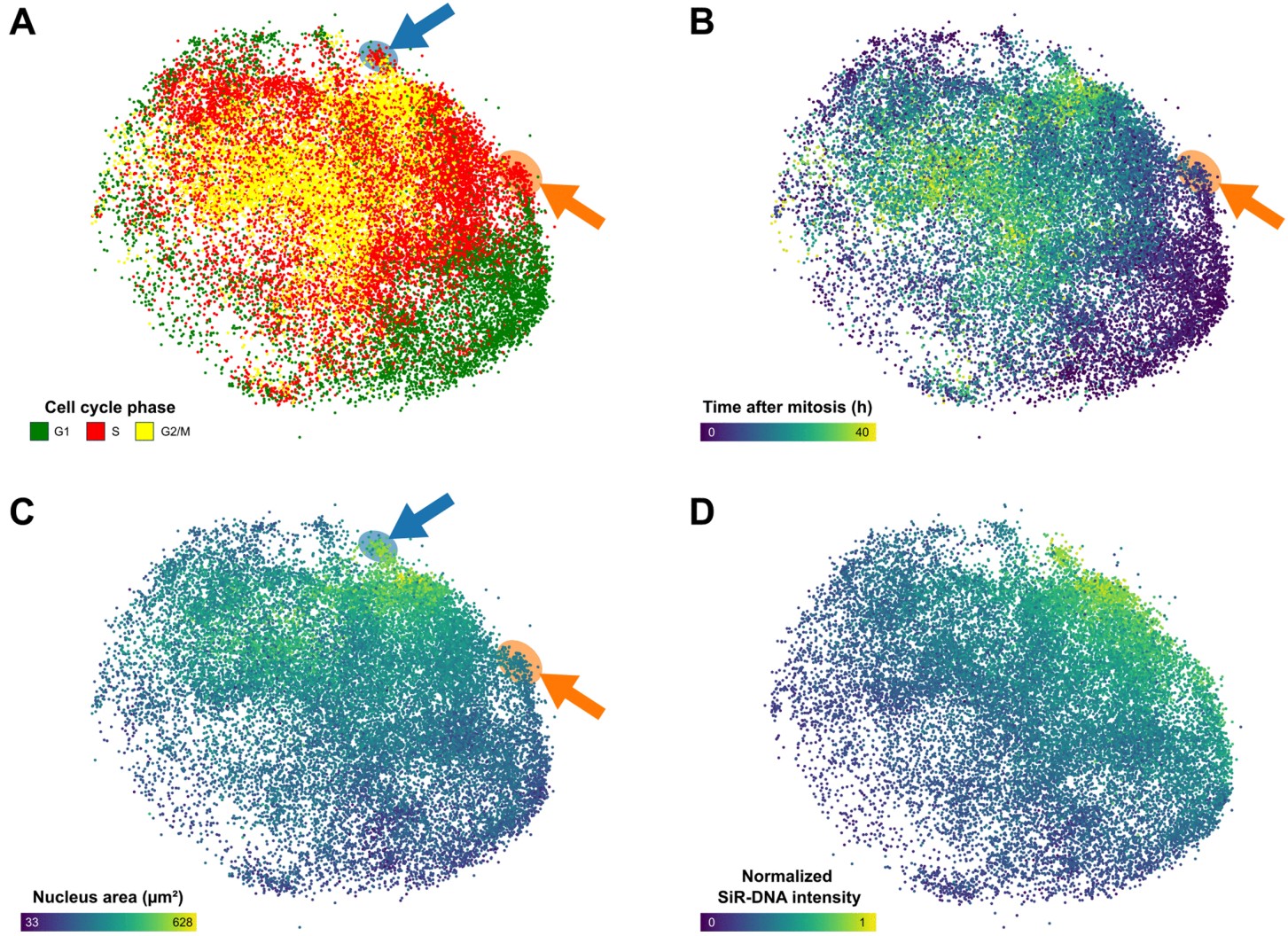

**Fig 4**. **UMAP representation of the nucleus embeddings learned by CC-VAE.** Each dot corresponds to a nucleus, colored by: (A) cell cycle phase (B) time elapsed since mitosis (C) area (D) normalized SiR-DNA intensity. The blue arrow points to large S-phase nuclei, and the orange arrow points to small early S-phase nuclei. These nuclei could be misclassified as G2/M or G1, respectively, emphasizing the necessity for a classification model that captures subtle phenotypic details beyond basic morphological features.

were misclassified as S. This supports the observations from Fig 4, indicating that simple morphological features such as nucleus area may not be sufficient to accurately determine the cell cycle phase of a nucleus. Finally, the unusual shape of the first nucleus in Fig 7D likely perturbed the classifier, as it exhibits a phenotype that is either absent or very rare in the training dataset.

Next, we compared the performance of our model to four alternative strategies for nucleus classification, inspired by related works mentioned above. As the models from these studies could not be used directly due to differences in input modality, we trained them on our dataset to allow for a fair comparison of their relative performances.

First, we trained a Support Vector Machine (SVM) classifier with linear kernel that took as input the nucleus's area and total SiR-DNA intensity. This baseline corresponds to what was traditionally used as a cell cycle proxy, for example in [26]. Next, we trained a CNN with the same backbone as CC-VAE and pretrained on ImageNet (a dataset of natural

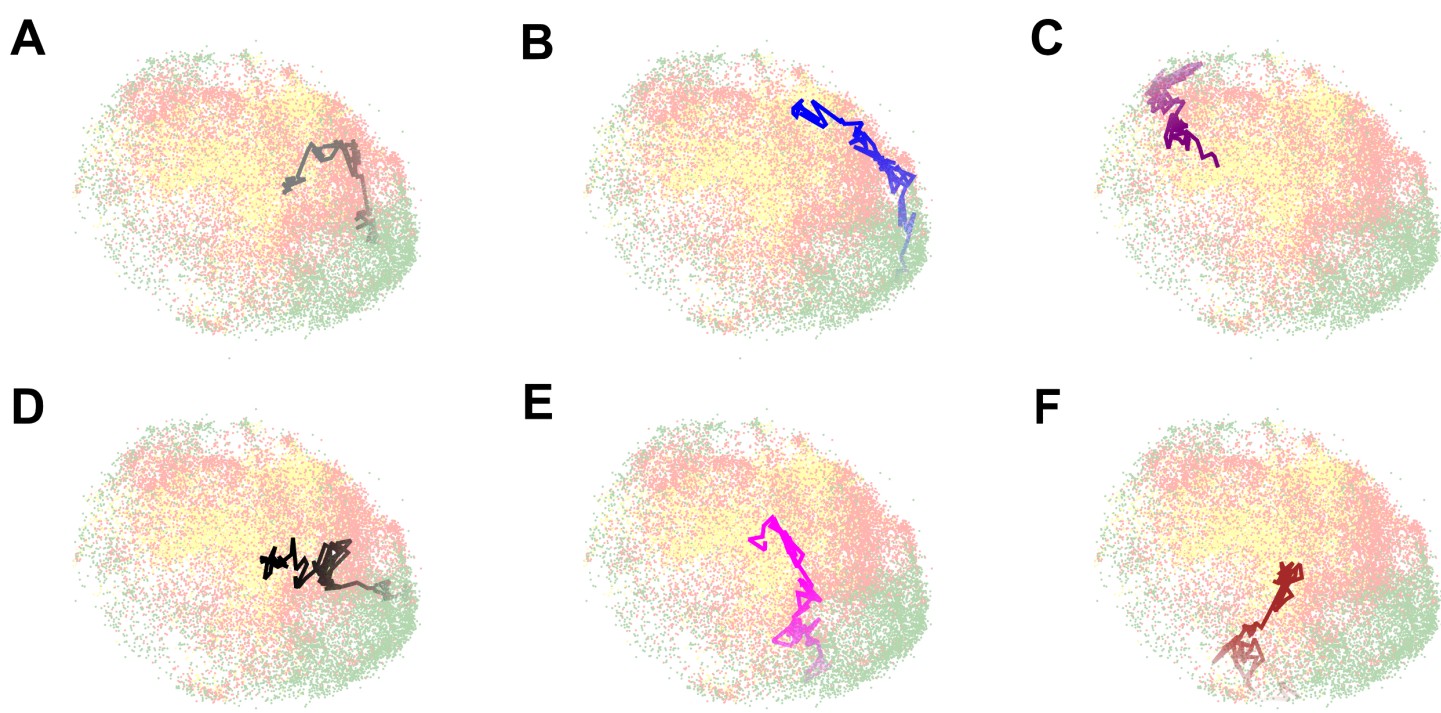

**Fig 5**. **Nucleus trajectories in the UMAP representation of the CC-VAE latent space.** Each trajectory follows a nucleus over time, from the G1 phase to the G2/M phase. The early steps of the trajectory are rendered transparent.

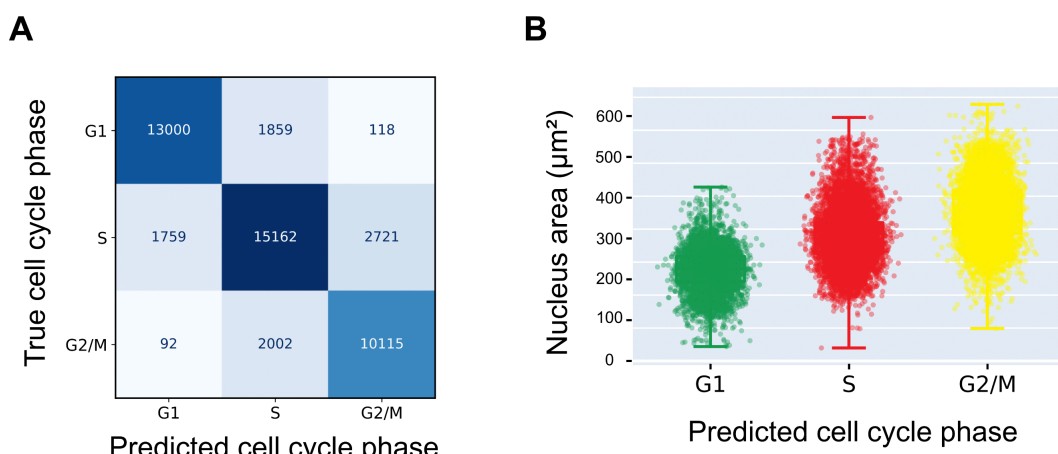

**Fig 6**. **Classification results.** (A) Average confusion matrix from 10-fold cross-validation of CC-VAE. (B) Nucleus area (µm²) as a function of the predicted cell cycle phase. Each dot represents a nucleus from the same subset of the test set shown in Fig 3.

images), as used in [22–25]. Note that all CNN weights were trainable during the classification training, whereas our CC-VAE approach only trains the last two fully connected layers on top of the frozen features. Finally, we implemented two state-of-the-art discriminative self-supervised learning strategies: SimCLR [38] and BYOL [39]. SimCLR employs a contrastive learning framework, while BYOL uses a non-contrastive approach based on self-distillation, similar to the method implemented in [19].

PLOS Computational Biology

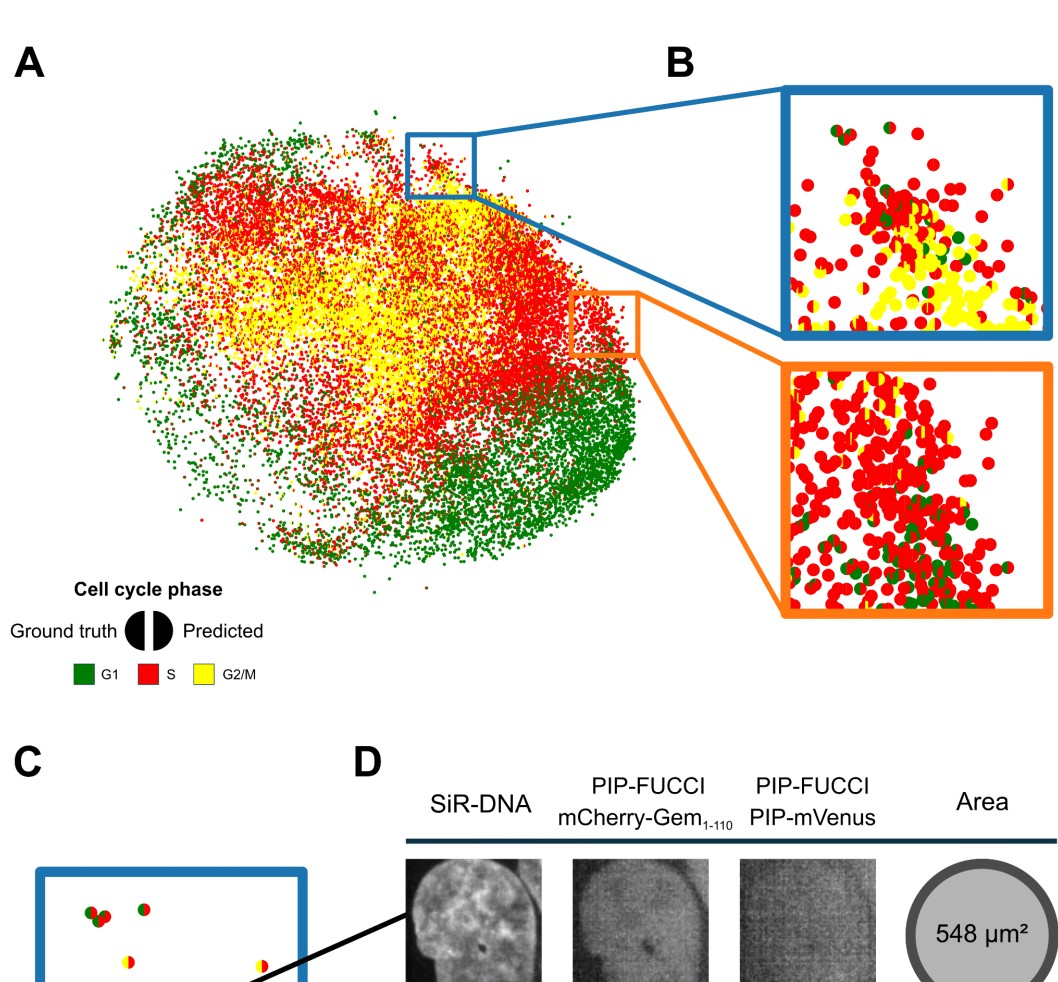

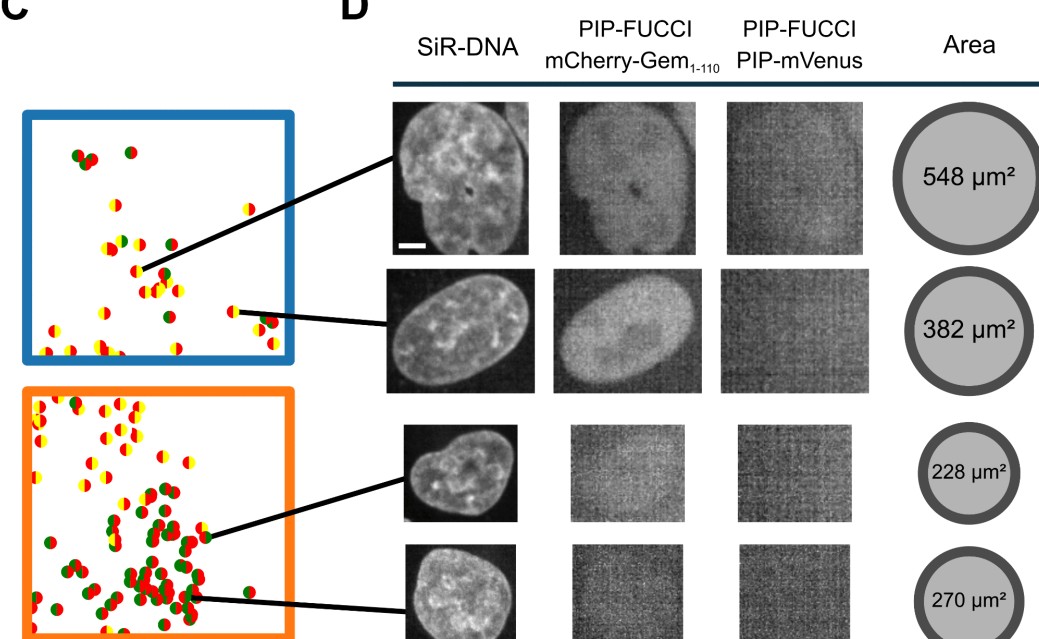

**Fig 7. Exploring the UMAP representation of nucleus embeddings learned by CC-VAE.** (A) Each dot represents a nucleus, colored by the ground truth and CC-VAE-predicted cell cycle phase on the left and right halves, respectively. (B) Zoom on phase-transition regions: G1 to S (orange) and S to G2/M (blue). (C) Classification errors observed in these phase-transition regions. (D) Misclassification examples: the first two rows show large S-phase nuclei misclassified as G2/M, while the last two rows show small early S-phase nuclei incorrectly labeled as G1. Scale bar: 5μm.

The methods were evaluated using two different metrics.

First, we evaluated time consistency using the top-1 retrieval accuracy, adapted from [40]. This metric assesses the model's ability to accurately retrieve positive pairs (consecutive frames) within a batch, reflecting the temporal coherence of the learned representations. Specifically, for a positive pair $(i,j)$ in a batch, the retrieval is considered successful if $j = \underset{k \neq i}{\arg\min} \|\mathbf{z}_i - \mathbf{z}_k\|$, meaning that the closest representation to $i$ within the batch (in terms of Euclidean distance) is $j$. The final score is obtained by averaging this result across all positive pairs of all elements in the test set.

Second, we evaluated the cell cycle phase classification performance using macro-averaged accuracy. This metric ensures fair handling of class imbalance by averaging accuracy across all classes. Results are displayed in Table 1.

We observe that the CNN trained from scratch slightly outperforms CC-VAE ($0.836 \pm 0.008$ vs $0.824 \pm 0.004$). However, this approach significantly underperforms in terms of time consistency, as it struggles to achieve the high top-1 retrieval accuracy of CC-VAE ($0.805 \pm 0.003$ vs $0.989 \pm 0.000$). This quantitative difference in performance is further illustrated by the trajectories of individual nuclei throughout the cell cycle. S1 Fig displays the nucleus trajectories in the CNN latent space, for the same tracks shown in Fig 5. While CC-VAE shows consistent trajectories that allow for smooth tracking of nuclei throughout the cycle, the CNN exhibits highly discontinuous and unstable trajectories, with large oscillations from one extreme of the latent space to the other. Such a lack of temporal consistency makes the CNN representations less reliable and less aligned with biological reality, as they fail to capture the continuous progression of the cell cycle.

## 2.5 Ablation study

We conducted an ablation study to investigate the impact of the different algorithmic elements. This study compares the performance of different models by successively removing specific elements from the CC-VAE method.

All models share the same backbone architecture. After potential pretraining on a pretext task, the encoder weights are frozen, and only the final two fully connected layers are trained for supervised cell cycle phase classification.

The first model is not pretrained on any task, and its weights are therefore randomly initialized. Next, we evaluated a model pretrained on the standard ImageNet classification task. We then tested vanilla Auto-Encoder (AE) and $\beta$-VAE, followed by our CC-VAE trained to predict PIP-FUCCI mean intensity as secondary task ($\mathcal{L} = \mathcal{L}_{\beta\text{-VAE}} + \lambda_1 \cdot \mathcal{L}_{\text{PIP-FUCCI}}$). Finally, we assessed our complete model, CC-VAE, with $\mathcal{L}_{\text{CC-VAE}} = \mathcal{L}_{\beta\text{-VAE}} + \lambda_1 \cdot \mathcal{L}_{\text{PIP-FUCCI}} + \lambda_2 \cdot \mathcal{L}_{\text{CONTRASTIVE}}$.

Additionally to the two metrics presented in the benchmarking, for VAE models, the reconstruction error was assessed using the Structural SIMimlarity (SSIM) metric. SSIM focuses on structural information rather than pixel-wise differences, making it particularly suitable for microscopy image analysis compared to traditional metrics like Mean Squared Error (MSE). Results are displayed in Table 2.

The AE achieves the best performance in reconstruction, which is expected since the model does not have to perform the additional tasks of CC-VAE, leading to fewer constraints on the model. While the two CC-VAEs perform quite similarly in terms of classification accuracy, the addition of the time regularization term in the loss enhances the time consistency of the latent space. As mentioned above, temporal consistency is quantified by the top-1 retrieval accuracy, defined as

Table 1. **Benchmarking results.** Bold entries highlight the best score for each task. Although the CNN achieves the best classification performance, it significantly lags behind CC-VAE in terms of time consistency.

| Model | Top-1 retrieval accuracy ↑ | Classification accuracy ↑ |
|---|---|---|
| SVM | - | $0.591 \pm 0.006$ |
| CNN | $0.805 \pm 0.003$ | $\mathbf{0.836 \pm 0.008}$ |
| SimCLR | $0.885 \pm 0.005$ | $0.797 \pm 0.014$ |
| BYOL | $0.706 \pm 0.125$ | $0.790 \pm 0.007$ |
| CC-VAE | $\mathbf{0.989 \pm 0.000}$ | $0.824 \pm 0.004$ |

**Table 2. Ablation study.** Bold entries highlight the best score for each task. While the model pretrained with an AE achieves the best reconstruction, our proposed method CC-VAE outperforms other methods both on time regularization and cell cycle classification.

| Pretext task | Reconstruction SSIM ↑ | Top-1 retrieval accuracy ↑ | Classification accuracy ↑ |
|---|---|---|---|
| None | - | $0.911 \pm 0.022$ | $0.526 \pm 0.008$ |
| ImageNet classification | - | $0.790 \pm 0.002$ | $0.635 \pm 0.004$ |
| AE | $\mathbf{0.659 \pm 0.004}$ | $0.957 \pm 0.001$ | $0.685 \pm 0.008$ |
| $\beta$-VAE | $0.655 \pm 0.004$ | $0.957 \pm 0.001$ | $0.697 \pm 0.008$ |
| CC-VAE, $\lambda_2 = 0$ | $0.639 \pm 0.003$ | $0.963 \pm 0.001$ | $0.819 \pm 0.003$ |
| CC-VAE | $0.633 \pm 0.003$ | $\mathbf{0.989 \pm 0.000}$ | $\mathbf{0.824 \pm 0.004}$ |

the percentage of cases in which a cell's representation in one frame is closest to its own representation in the consecutive frame among all cells. Notably, we observe that ImageNet pretraining reduces this temporal consistency compared to an untrained network. The untrained network yields relatively high values, as the representations of the same cell in consecutive frames are naturally very similar. Focusing solely on discrimination—especially when pretrained on a different domain—can thus disrupt this intrinsic similarity, which further supports the use of explicit temporal regularization in our approach.

### 2.6 Inference on unlabeled nuclei

We performed inference on the 346,028 nuclei that remained unlabeled due to atypical PIP-FUCCI intensity variations, to verify that no distributional shift exists between labeled and unlabeled nuclei.

Fig 8 shows UMAP representations of the nucleus embeddings learned by CC-VAE on a subset of the unlabeled nuclei. This subset was randomly sampled from the 346,028 unlabeled nuclei to match the number of nuclei displayed in Fig 3, thereby enabling a fair comparison.

Interestingly, a substantial number of nuclei are located in the lower-left region of the UMAP. These correspond to early-G1 nuclei, which can be readily identified as the nucleus has only just formed, resulting in a small size, weak SiR-DNA signal and no detectable PIP-FUCCI expression. The lack of sufficient SiR-DNA signal may have caused tracking failures, explaining why these nuclei were disregarded. In other cases, the absence of clear PIP-FUCCI transitions made ground truth annotation ambiguous, and such nuclei were therefore excluded rather than incorrectly labeled. Although ground truth annotation is missing, Fig 8B indicates that our model can still reliably classify these nuclei, demonstrating its ability to infer the cell cycle phase on unseen data. Overall, more than half of the unlabeled nuclei are predicted as G1 (52.9% G1, 23.7% S and 23.4% G2/M).

More generally, Fig 8 shows that the unlabeled nuclei are distributed across the entire embedding space. They span different phases of the cell cycle and were excluded only because their PIP-FUCCI intensity changes did not allow for a reliable determination of phase boundaries: for example, when tracks were too short, covered only a single phase, or were truncated at the end of the video.

## 3 Discussion

We present Cell Cycle Variational Auto-Encoder (CC-VAE), a novel deep self-supervised method for determining the cycle phase of individual cells based on a DNA fluorescence marker. CC-VAE can serve as a powerful tool for cell stratification according to their cycle phase, enabling for example the study of new connections between the cell cycle and patterns of RNA or protein localization.

For this, we propose variational auto-encoders, that provide generic and powerful latent space representations. We introduce two auxiliary tasks, prediction of the cell-cycle dependent PIP-FUCCI fluorescent intensities and a novel time-consistency constraint. Together, they structure the latent space to group nuclei according to their cell cycle phase while preserving phenotypic continuity ensuring that nearby nuclei in the latent space are also phenotypically close. A

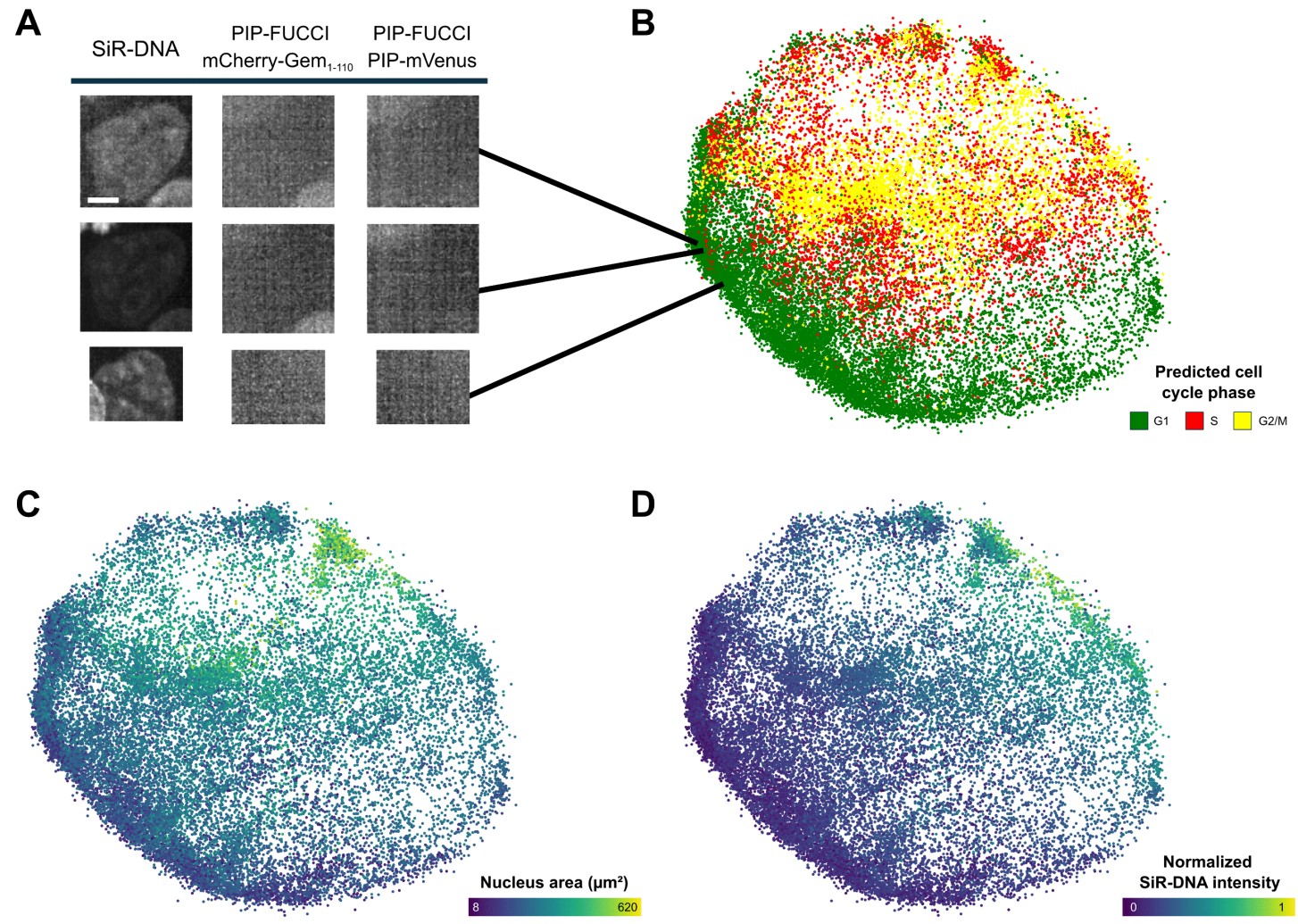

**Fig 8**. **Unlabeled nuclei.** (A) Unlabeled early-G1 nuclei. Scale bar: 5μm. (B-D) UMAP representations of the unlabeled nucleus embeddings. Each dot corresponds to a nucleus, colored by: (B) predicted cell cycle phase (C) area (D) normalized SiR-DNA intensity.

lightweight classifier trained on the frozen latent representations achieves an average classification accuracy of 82.4%. All code and the trained networks are made freely available at GitHub, https://github.com/15bonte/cell_cycle_classification and HuggingFace, https://huggingface.co/thomas-bonte/cell_cycle_classification, respectively.

Notably, our strategy optimizes relatively few parameters specifically for classification. While training a full CNN end-to-end for classification can yield slightly higher accuracy, we show that this comes at the cost of temporal and phenotypic consistency. Temporal consistency is particularly important, as it reflects the inherently continuous nature of the cell cycle, even though we discretize it into three main phases in this work. This discretization should be interpreted with caution, as it represents a simplified model of a fundamentally continuous biological process. Therefore, we argue that the biological plausibility of our approach should not only be viewed as a means to maximize classification accuracy, but as an intrinsic strength of the method itself. We believe that maintaining biological consistency is essential, even if it comes at the cost of a slight reduction in classification performance. Variational auto-encoders provide a natural mechanism to add biologically

meaningful pretext tasks and thus impose a biological structure to the latent space. On the contrary, relying on latent representations that do not preserve biological distances can be misleading, as they might suggest phenotypic discontinuities that do not correspond to a biological reality. Additionally, optimizing solely for classification accuracy risks overfitting to a specific dataset, whereas our goal is to develop a robust method suitable for broader application.

Another important aspect of the study is the overall annotation strategy. We found that tracking PIP-FUCCI over time is essential to reliably identify cell cycle phases due to high variability in marker intensity between individual cells. While the intensity time series allow for a reliable cell cycle assessment, static snapshots may lead to label noise that propagates through the learning process. Here, we have carefully annotated a large number of cells over time, leading to a dataset of over 600,000 annotated nuclei. This dataset is made freely available on BioImage Archive at https://www.ebi.ac.uk/biostudies/bioimages/studies/S-BIAD1659 to support further community development and benchmarking.

Our current approach is not free of limitations. First, we can expect some improvements by incorporating additional label-free modalities, such as bright-field or phase-contrast imaging, providing complementary phenotypic information missing from DNA fluorescence markers, and thus potentially improving both cell cycle phase prediction and overall phenotypic consistency of the latent space. This enhancement would be particularly valuable for biologists as these modalities are common, label-free, and already frequently used in daily experiments. Furthermore, the current approach is still vulnerable to domain shifts induced by the use of other DNA markers, thus limiting direct out-of-the-box use of the trained networks. However, since SiR-DNA and Hoechst are derived from the same molecule [29], and both SiR-DNA and DAPI intercalate into AT-rich regions of DNA [29,41], it may be possible to use domain adaptation techniques to adapt our networks to these markers. This adaptation could help democratize CC-VAE, making it a more accessible and practical tool for biologists. Finally, CC-VAE enhances the $\beta$-VAE architecture by integrating in-silico labeling and time-contrastive pretext tasks, in order to structure the latent space for cell cycle disentanglement. However, alternative pretext tasks could have been explored. Notably, Fig 4B and S2 Fig suggest – as expected – a strong correlation between cell cycle phases and the time elapsed after mitosis. Predicting this temporal information could serve as a powerful pretext task for cell cycle phase classification. Yet, this approach is not feasible in our case, as most tracks do not span the entire cell cycle but only partial segments. As a result, the exact onset of the cycle remains unknown for most nuclei, preventing access to precise elapsed time since mitosis. However, this method could be worth exploring on a data set composed exclusively of tracks that span the entire cell cycle.

In summary, here we introduced CC-VAE, a powerful self-supervised-based method for accurately predicting the cell cycle phase from DNA marker microscopy images. It improves the $\beta$-VAE architecture by incorporating in-silico labeling and time-contrastive additional tasks, enabling the model to capture subtle phenotypic information related to cell cycle progression. Leveraging this rich latent encoding, CC-VAE achieves high accuracy and robustness in cell cycle phase classification. Additionally, we provide a novel dataset containing over 600,000 labeled nucleus images, which we believe will be a valuable resource for ongoing and future research in the field.

## 4 Methods

### 4.1 Variational Auto-Encoders

Variational Auto-Encoders (VAE, [33]) belong to the family of self-supervised generative models. Compared to vanilla auto-encoders, they provide better latent representations by introducing regularization, both at the local and global level. In the following, bold symbols (e.g., $\mathbf{x}$) denote vector quantities.

Let $\mathbf{x} \in \mathbb{R}^D$ be a set of observable variables deriving from an unknown distribution. VAE assume that there exist latent variables $\mathbf{z} \in \mathbb{R}^d$, $\mathbf{z}$ being a latent representation of $\mathbf{x}$ with $D \gg d$, i.e. the latent space is of much smaller dimension than the original space. The generative model writes $\mathbf{z} \sim p_{\mathbf{z}}(\mathbf{z})$ ; $\mathbf{x} \sim p_{\theta}(\mathbf{x}|\mathbf{z})$, with $p_{\theta}(\mathbf{x}|\mathbf{z})$ taken as a parametric distribution.

Therefore, the marginal likelihood $p_\theta(\mathbf{x})$ is given by:

$$p_\theta(\mathbf{x}) = \int_{\mathbb{R}^d} p_\theta(\mathbf{x}|\mathbf{z})p_{\mathbf{z}}(\mathbf{z})d\mathbf{z} \tag{4}$$

Since the integral is taken over the entire latent space, it is most of the time computationally intractable – meaning that the computation complexity of any approach for evaluating this integral is exponential. So does the posterior distribution $p_\theta(\mathbf{z}|\mathbf{x})$, which is then approximated by a parametric distribution $q_\phi(\mathbf{z}|\mathbf{x})$ through variational inference [42]. $q_\phi(\mathbf{z}|\mathbf{x})$ and $p_\theta(\mathbf{x}|\mathbf{z})$ are refered to as the *encoder* and the *decoder*, respectively, whose parameters are given by deep neural networks. Importance sampling enables to define an unbiased estimate $\hat{p}_\theta(\mathbf{x}) = \frac{p_\theta(\mathbf{x},\mathbf{z})}{q_\phi(\mathbf{z}|\mathbf{x})}$ of the marginal likelihood $p_\theta(\mathbf{x})$, and to obtain a lower bound for the marginal log-likelihood through Jensen's inequality:

$$\log p_\theta(\mathbf{x}) \geq \mathbb{E}_{\mathbf{z}\sim q_\phi}[\log p_\theta(\mathbf{x}|\mathbf{z})] - \mathrm{KL}(q_\phi(\mathbf{z}|\mathbf{x})\|p_{\mathbf{z}}(\mathbf{z})) := \mathcal{L}(\theta, \phi, \mathbf{x}) \tag{5}$$

$\mathcal{L}(\theta, \phi, \mathbf{x})$ is called the Evidence Lower BOund (ELBO). Since we aim at maximizing the marginal log-likelihood $\log p_\theta(\mathbf{x})$ (as the probability of the model itself), the ELBO can be considered as a good proxy to maximize during training. In vanilla VAEs, the prior $p_{\mathbf{z}}(\mathbf{z})$ is chosen as a standard Gaussian distribution $\mathcal{N}(\mathbf{0}, \mathbf{I_d})$.

## 4.2 Live cell imaging

Hela Kyoto cells stably expressing Cdt1(1-17aa)-HA-mVenus and mCherry-Geminin(1-110aa) were cultured in 10 cm dishes in DMEM + 10%FBS + 1% PenStep at 37°C with 5% $CO_2$. For imaging, cells were cultured on a 96-well phenoplate (Revvity). Culture media was aspirated and cells were incubated with 100 nm SiR-DNA Cy5 (Spirochrome) for 4h at 37°C. The media was then replaced with imaging media pre-warmed at 37°C (DMEM fluorobrite, 1% P/S, 10% FBS, glutamax, 100 nm SiR DNA) at least 2h before imaging.

One day after seeding, time-lapse microscopy was performed via live-cell fluorescence imaging using an Opera Phenix Plus High-Content Screening System (Perkin Elmer) with a 63X water immersion objective (1.15 NA). Imaging lasted for 40 hours, capturing images every 20 minutes. A total of 159 fields-of-view were acquired, with 5 focal planes per field spaced 1 µm apart. 3 channels (488 nm, 561 nm and 640 nm) were acquired, each with 20% laser power and 100 ms exposure time. The temperature and $CO_2$ level were maintained at 37°C and 5%, respectively. One pixel corresponds to 0.102 µm.

## 4.3 Image preprocessing

Nucleus segmentation was performed on each frame independently, using Cellpose [43] applied to the SiR-DNA channel. We used the Cellpose interface to conduct human-in-the-loop fine-tuning of the default Cellpose segmentation model. This involved using 20 frames randomly selected from our dataset. We retained Cellpose's default training parameters: a learning rate of 0.1, weight decay of 0.0001, and 100 epochs. Segmentation was performed with the flow threshold set to 0.4 and nucleus probability threshold set to 0. The nucleus diameter is automatically calculated by Cellpose and set to 173.90 px, i.e. 17.74 µm.

The tracking is performed using the particle tracking algorithm developed by [44] and implemented in TrackMate [45]. The maximum distance for tracking is set to 200 px, i.e. 20.4 µm. Track merging, track splitting and gap closing are not permitted. Across the dataset, a total of 15,026 nucleus tracks were identified, encompassing 982,332 single nucleus images.

The nuclei were labeled on the basis of the temporal evolution of their PIP-FUCCI intensities across their area. 217,465 nuclei were labeled as G1, 257,984 were labeled as S, and 160,855 were labeled as G2/M. Additionally, 346,028 nuclei remained unlabeled.

All Trackmate tracks were manually verified during this cell cycle phase annotation. Although the annotation was partially automated, each of the 15,026 tracks was manually reviewed to correct potential errors. This process allowed us not only to refine phase annotations but also to identify tracking errors by detecting inconsistencies in the PIP-FUCCI signals. Overall, however, tracking quality was excellent, owing to the high temporal resolution and the strong signal-to-noise ratio provided by SiR-DNA.

No maximum projection was applied prior to training; instead, all z-stack planes were used as input to the model. SiR-DNA images were normalized before being provided as input to the models.

## 4.4 Deep learning training

**4.4.1 Cross-validation.** The 159 fields-of-view in our dataset are divided into 10-fold cross-validation sets, ensuring that nuclei from the same field-of-view are grouped within the same set. Each fold includes 80% of the data for training, 10% for validation, and the same 10% for testing.

**4.4.2 Model.** Our encoder is a standard ResNet18 with 11.4M parameters. The decoder, designed to mirror the encoder's structure, is symmetrical. The self-supervised model was trained for 10 epochs with a learning rate of 1e-4 and a batch size of 128. As the training curves plateaued after 10 epochs, extending training beyond this point was unnecessary. One epoch takes on average 2 hours on a P100 GPU. We used data augmentations including rotation, horizontal flip, and vertical flip. Early stopping was employed to select the best model based on validation performance. The implementation of the VAE-based models relies on Pythae [46].

The average intensities of PIP-FUCCI are predicted using a single linear layer applied to the latent representation $\mathbf{z}$. The contrastive NT-Xent loss is applied in a manner similar to the approach described in [38], with the key difference being that the only augmentation introduced in our method is a one-frame shift. We retained the default temperature parameter value of 0.7.

For supervised classification training, we train two fully connected layers (67k parameters) on top of the frozen encoder features to classify the nucleus images into one of three phases: G1, S, or G2/M. Training was performed during 1 epoch using a learning rate of 1e-4 and a batch size set to 64, with the cross-entropy loss function used to optimize classification accuracy. While training deep learning models for only a single epoch is uncommon, in this case we trained only the 67k parameters of the classifier on a dataset of over 500k nucleus images. This resulted in very rapid convergence of the loss curves, rendering additional epochs unnecessary.

**4.4.3 Selection of hyperparameters.** To improve classification accuracy, we performed a grid search to select optimal values for $\beta$, the self-supervised learning rate, the latent dimension, $\lambda_1$ and $\lambda_2$. To limit training time, this grid search was performed on 5% of the train and validation data. Additionally, to reduce the number of combinations, we first considered the three parameters used in the standard $\beta$-VAE ($\beta$, learning rate, latent dimension), followed by $\lambda_1$, and finally $\lambda_2$.

$\beta$ was chosen in $\{0.01, 0.1, 1\}$ as [46] suggests that selecting $\beta > 1$ can negatively impact classification accuracy. The learning rate was chosen in $\{5\text{e-}5, 1\text{e-}4, 5\text{e-}4\}$ and the latent dimension was chosen in $\{64, 128, 256, 512\}$. Table 3 reports the classification accuracy obtained for different combinations of the three hyperparameters. Smaller values of $\beta$ lead to better classification performance, consistent with the findings of [46] on other imaging modalities. Overall, larger latent spaces improve classification accuracy, as they allow the model to capture more subtle phenotypic details of the nuclei. However, this improvement plateaus, with latent dimensions of 256 and 512 yielding comparable results. The optimal hyperparameters were finally found to be $\beta = 0.01$, a latent dimension of 256 and a learning rate of 1e-4.

$\lambda_1$ and $\lambda_2$ are chosen in $\{10, 100, 1\text{e3}, 1\text{e4}, 1\text{e5}\}$ and $\{10, 100, 1\text{e3}\}$, respectively. These sets were chosen based on the initial ratio between the $\beta$-VAE loss and the additional losses. Table 4 reports the classification accuracy obtained for different values of these hyperparameters. The optimal values were found to be $\lambda_1 = 1\text{e4}$ and $\lambda_2 = 100$.

**4.4.4 Training of the benchmark models.** The CNN presented in Table 1 is a ResNet18 pretrained on ImageNet classification. Classification training was performed during 10 epochs using a learning rate of 1e-4, a batch size set to 64.

**Table 3. Hyperparameter search for $\beta$-VAE.** $\beta$-VAE classification accuracy results for different values of the hyperparameters $\beta$, latent dimension ($ld$), and learning rate ($lr$). Note that these results are substantially lower than those reported in Table 1, as the models were trained on only 5% of the data to reduce training time. The optimal parameters were found to be $\beta$=0.01, $ld$=256, and $lr$=1e-4.

| | | $lr$=5e-5 | 1e-4 | 5e-4 |
|---|---|---|---|---|
| $\beta$=0.01 | $ld$=64 | 0.625 ± 0.015 | 0.629 ± 0.016 | 0.628 ± 0.014 |
| | 128 | 0.650 ± 0.014 | 0.651 ± 0.015 | 0.648 ± 0.012 |
| | 256 | 0.661 ± 0.015 | **0.662 ± 0.011** | 0.656 ± 0.015 |
| | 512 | 0.661 ± 0.014 | 0.658 ± 0.013 | 0.654 ± 0.015 |
| 0.1 | 64 | 0.622 ± 0.011 | 0.616 ± 0.011 | 0.620 ± 0.020 |
| | 128 | 0.649 ± 0.013 | 0.644 ± 0.009 | 0.647 ± 0.012 |
| | 256 | 0.656 ± 0.012 | 0.655 ± 0.012 | 0.654 ± 0.018 |
| | 512 | 0.654 ± 0.017 | 0.660 ± 0.016 | 0.647 ± 0.010 |
| 1 | 64 | 0.586 ± 0.013 | 0.590 ± 0.015 | 0.586 ± 0.013 |
| | 128 | 0.604 ± 0.011 | 0.604 ± 0.014 | 0.599 ± 0.013 |
| | 256 | 0.610 ± 0.014 | 0.613 ± 0.017 | 0.611 ± 0.014 |
| | 512 | 0.611 ± 0.014 | 0.616 ± 0.018 | 0.613 ± 0.014 |

**Table 4. Hyperparameter search for $\lambda_1$ and $\lambda_2$.** CC-VAE classification accuracy results for different values of the hyperparameters $\lambda_1$ and $\lambda_2$. Note that these results are substantially lower than those reported in Table 1, as the models were trained on only 5% of the data to reduce training time. The optimal parameters were found to be $\lambda_1$=1e4 and $\lambda_2$=100.

| $\lambda_1$=10 | 100 | 1e3 | 1e4 | 1e5 |
|---|---|---|---|---|
| 0.662 ± 0.013 | 0.676 ± 0.013 | 0.725 ± 0.011 | **0.758 ± 0.010** | 0.757 ± 0.013 |
| | $\lambda_2$ = 10 | 100 | 1e3 | |
| | 0.764 ± 0.011 | **0.768 ± 0.008** | 0.759 ± 0.01 | |

We used data augmentations including rotation, horizontal flip, and vertical flip. Early stopping was employed to select the best model based on validation performance.

SimCLR was trained for 10 epochs with a learning rate of 1e-4 and a batch size of 128. Standard values were used for the LARS optimizer: momentum of 0.9, weight decay of 1e-6, and epsilon of 1e-5. The number of views was set to 2, and the temperature to 0.07. We used data augmentations including random cropping, horizontal and vertical flips, color jittering, and Gaussian blur. To match the format of standard RGB images required for color jittering, only the 3 central z-stacks (out of 5) were used as input. Early stopping was employed to select the best model based on validation performance. For supervised classification, we followed the same procedure as for CC-VAE.

BYOL was trained for 10 epochs with a learning rate of 1e-4 and a batch size of 128. We used data augmentations including color jittering, grayscale conversion, horizontal and vertical flips, Gaussian blur, and random cropping. To match the format of standard RGB images required for color jittering, only the 3 central z-stacks (out of 5) were used as input. Early stopping was employed to select the best model based on validation performance. For supervised classification, we followed the same procedure as for CC-VAE.

## Supporting information

**S1 Fig. Nucleus trajectories in the UMAP representation of the CNN latent space.** (A–F) Displayed tracks originate from the same cells as those in Fig 5. The early steps of the trajectory are rendered transparent. (G) UMAP representation of the CNN latent space.
(PDF)

**S2 Fig. Time elapsed after mitosis (hours).** Each dot represents a nucleus from the same subset of the test set shown in Fig 3.
(PDF)

## Author contributions

**Conceptualization:** Thomas Bonte, Adham Safieddine, Florian Muller, Dominique Weil, Edouard Bertrand, Thomas Walter.

**Data curation:** Thomas Bonte, Oriane Pourcelot, Adham Safieddine.

**Formal analysis:** Thomas Bonte.

**Funding acquisition:** Edouard Bertrand, Thomas Walter.

**Investigation:** Thomas Bonte, Edouard Bertrand, Thomas Walter.

**Methodology:** Thomas Bonte, Adham Safieddine, Floric Slimani, Dominique Weil, Edouard Bertrand, Thomas Walter.

**Project administration:** Thomas Walter.

**Resources:** Oriane Pourcelot, Edouard Bertrand.

**Software:** Thomas Bonte.

**Supervision:** Edouard Bertrand, Thomas Walter.

**Validation:** Thomas Bonte, Edouard Bertrand, Thomas Walter.

**Visualization:** Thomas Bonte.

**Writing – original draft:** Thomas Bonte, Thomas Walter.

**Writing – review & editing:** Thomas Bonte, Oriane Pourcelot, Adham Safieddine, Floric Slimani, Edouard Bertrand, Thomas Walter.

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
