## [Decision Letter · Decision Letter 0]

31 Aug 2025

PCOMPBIOL-D-25-01257

A Deep Learning approach for time-consistent cell cycle phase prediction from microscopy data

PLOS Computational Biology

Dear Dr. Walter,

Thank you for submitting your manuscript to PLOS Computational Biology. After careful consideration, we feel that it has merit but does not fully meet PLOS Computational Biology's publication criteria as it currently stands. Therefore, we invite you to submit a revised version of the manuscript that addresses the points raised during the review process.

Please submit your revised manuscript within 60 days Oct 31 2025 11:59PM. If you will need more time than this to complete your revisions, please reply to this message or contact the journal office at ploscompbiol@plos.org. Please include the following items when submitting your revised manuscript:

We look forward to receiving your revised manuscript.

Kind regards,

Virginie Uhlmann

Academic Editor

PLOS Computational Biology

Stacey Finley

Section Editor

PLOS Computational Biology

**Additional Editor Comments:**

Reviewer #1:

Reviewer #2:

Reviewer #3:

Reviewer #4:

**Journal Requirements:**

- TM on page: 14.

6) Kindly revise your competing statement to align with the journal's style guidelines: 'The authors declare that there are no competing interests.'

7) Kindly update the description of the supplementary file in the file inventory from 'cover letter' to 'Supplementary Fig 1.

**Reviewers' comments:**

Reviewer's Responses to Questions

**Comments to the Authors:**

Reviewer #1: The authors of “A Deep Learning approach for time-consistent cell cycle phase prediction from microscopy data” present an interesting framework (CC-VAE) for cell cycle phase prediction using several novel approaches such as contrastative learning and a clever formulation of the loss function. The manuscript is well-written and easy to follow, and I think CC-VAE would definitely be a valuable tool for the community.

However, I have two major points:

1. The benchmarking should also include one of the methods described in the introduction. I recommend comparing to “Ulicna et al” (reference 20) on the same dataset used for CC-VAE

2. A proof-of-concept biological validation is missing. From the classified cell cycle phase on unseen data the authors should replicate a plot from another publication where cell cycle phase was used to obtain biological insights.

Other comments:

3. The authors point out that the method in “Ulicna et al” requires timelapse microscopy while their method does not. However, for the contrastative learning approach, timelapse is required right? Please explain better how CC-VAE can be used with snapshot data.

4. Along the lines of point 3, the authors seem to infer that CC-VAE has the advantage that is can be used with “standard fixed-cell microscopy data”. If true, it needs to be tested on such data.

5. While I like the idea of the ablation study in Table 2, I am confused about how can the model with randomly initialized weights have such a high Top-1 retrieval accuracy?? I think this needs more careful testing, maybe multiple random initializations to get an idea of the Top-1 retrieval accuracy distribution. Otherwise, it makes me doubt the usefulness of such a metric.

6. It is mentioned that tracking is performed with TrackMate. However, it is not clear whether the tracks were manually validated (or corrected) or not. Please clarify this. If not manually corrected, authors need to be very careful about tracking errors.

7. Unfortunately, I could not test the model because I got an ImportError when running the “eval_classifier” notebook. I included the traceback error below

8. The overall usability should be improved. A napari plugin would be the best, but the notebooks folder on the GitHub repo should include one notebook on how to run the classification from a single image file (possibily a TIFF file, included in the repo).

ModuleNotFoundError Traceback (most recent call last)

Cell In[1], line 12

9 from cnn_framework.utils.metrics.classification_accuracy import ClassificationAccuracy

11 from cell_cycle_classification.utils.data_set import FucciClassificationDataSet

- 12 from cell_cycle_classification.backbone.fucci_classifier import FucciClassifier

14 from cell_cycle_classification.utils.model_params import FucciVAEModelParams

File ~/cell_cycle_classification/cell_cycle_classification/backbone/fucci_classifier.py:5

1 """"Cell cycle classifier."""

3 import torch.nn as nn

-- 5 from .encoder_resnet import ResnetEncoder

6 from .model import FucciVAE

7 from ..utils.tools import get_final_model_path, get_vae_config

File ~/cell_cycle_classification/cell_cycle_classification/backbone/encoder_resnet.py:10

7 from pythae.models.nn import BaseEncoder

8 from pythae.models.base.base_utils import ModelOutput

- 10 from cnn_framework.utils.model_managers.utils.custom_get_encoder import get_encoder

13 def redefine_first_layer(model: nn.Module, input_channels: int) -> None:

14 """Redefine the first layer of the model to accept input_channels number of channels."""

File ~/.pyenv/versions/miniforge3-latest/envs/ccvae/lib/python3.10/site-packages/cnn_framework/utils/model_managers/utils/custom_get_encoder.py:14

12 from segmentation_models_pytorch.encoders.xception import xception_encoders

13 from segmentation_models_pytorch.encoders.timm_efficientnet import timm_efficientnet_encoders

- 14 from segmentation_models_pytorch.encoders.timm_resnest import timm_resnest_encoders

15 from segmentation_models_pytorch.encoders.timm_res2net import timm_res2net_encoders

16 from segmentation_models_pytorch.encoders.timm_regnet import timm_regnet_encoders

ModuleNotFoundError: No module named 'segmentation_models_pytorch.encoders.timm_resnest'

Reviewer #2: I enjoyed reading the paper and I congratulate you for this great work!

1. It is not clear if the VAE is really needed and what exactly it brings to the table (since generation is not the aim here). What would happen if you used a vanilla autoencoder instead? I think it would valuable to add an entry to the ablation experiment table (Table 2) where β=0.

2. Line 96: "46,028 nuclei remained unlabeled due to the inability to reliably determine cell cycle phase boundaries". Was the model tested on these unlabeled nuclei to determine if it still works? In other words is there a significant distribution shift between the unlabeled and labeled subsets?

Reviewer #3: Manuscript Number: PCOMPBIOL-D-25-01257

Title: A Deep Learning approach for time-consistent cell cycle phase prediction from microscopy data

Thank you for giving me the opportunity to review this article. The paper introduces CC-VAE, a β-Variational Auto-Encoder framework enhanced with in-silico labeling of phase-specific markers and temporal consistency regularization for cell cycle phase prediction from microscopy images stained with SiR-DNA. The authors also released a large annotated dataset (>600,000 HeLa Kyoto nuclei) along with trained models and code. The manuscript is well-written, technically sound, and reproducible, but I have several concerns that should be addressed before publication. Please find my comments below:

1. I am a bit concerned about the novelty and generalizability of this work. The integration of in-silico labeling and temporal consistency into a β-VAE is innovative, but the classification accuracy is slightly lower than that of a supervised CNN. The authors should clarify why CC-VAE is preferable when a simpler CNN performs better in accuracy, and under what conditions temporal coherence provides a practical advantage over raw performance.

2. The model is trained and evaluated solely on SiR-DNA–stained HeLa Kyoto cells from a single microscope, with no tests on other stains, cell types, or platforms. Including external datasets, cross-cell-line validation, or transfer-learning experiments would address generalization concerns and strengthen the claims of broader applicability.

3. The latent space visualizations are compelling, but misclassifications (particularly at G1/S and S/G2/M transitions) are not well explained in biological terms. An analysis linking errors to chromatin or morphological features would increase trust in the model among biological users.

4. The evaluation is limited to a morphology-based SVM and a supervised CNN. The authors should benchmark against recent self-supervised methods such as SimCLR, BYOL, MoCo, DINO, or masked autoencoders to better assess whether CC-VAE offers advantages over current deep learning approaches.

5. The introduction should provide a stronger motivation for the study by clearly stating the research gap it addresses.

6. The objectives of the research should be explicitly outlined at the end of the introduction.

7. At the end of the introduction, add the organization of the paper.

8. Format the paper correctly. The paper does not follow the standard PLOS Computational Biology structure. The Methods section comes after the Results and Discussion, making it hard for readers to understand the study before seeing the findings. Reorder sections as: Introduction → Methods → Results → Discussion → Conclusion.

9. The methodology section misses some important points.

a) How exactly are the temporal contrastive pairs selected in cases of irregular cell cycle timing?

b) Whether all z-stack planes were used or maximum projections were applied before training.

c) Normalization applied to the SiR-DNA intensity before input to the network.

10. Several figures could be improved for clarity:

a) In Fig. 4, arrows and cluster labeling in the UMAP could be made clearer for readers unfamiliar with embedding spaces.

b) In Fig. 5, adding a temporal color gradient would help visualize progression along the cell cycle trajectory.

11. While an ablation study is included, a more explicit sensitivity analysis (e.g., effect of β parameter, weight of temporal loss, or latent dimensionality) would improve transparency about the robustness of design choices.

Reviewer #4: The authors present the Cell Cycle Variational Auto-Encoder (CC-VAE), a novel self-supervised deep learning approach designed to determine the cell cycle phase of individual cells based on a DNA fluorescence marker. According to the manuscript, CC-VAE provides a powerful tool for stratifying cells by their cycle phase, thereby facilitating the investigation of potential links between the cell cycle and patterns of RNA or protein localization.

Temporal consistency is particularly important because the cell cycle is a continuous biological process, not a set of isolated events. Although for analytical convenience the cycle is often discretized into three main phases (G1, S, G2/M), in reality the transitions between them occur gradually, with overlapping molecular and morphological signatures. Ensuring temporal consistency in modelling therefore better reflects the underlying biology, prevents artificial discontinuities introduced by phase labeling, and improves the accuracy of downstream analyses such as tracking RNA or protein localization dynamics.

Overall, the authors have done a commendable job in preparing this manuscript - the research is solid, the results are clearly presented, and the language is well written; however, there are some minor issues that I would like them to address.

Fig. 1 shows that the G1 phase is represented by SiR-DNA and PIP-mVenus.

In the main text, the authors state: “Fig. 1 displays some example nuclei from our annotated dataset. G1 nuclei express only PIP-mVenus fluorescence, S nuclei express only mCherry-Gem1-110 fluorescence, and G2 nuclei express both signals” (lines 100–102).

Was there any difference in the testing of cells that expressed only SiR-DNA, only PIP-mVenus, or both together, if such alone-expression is possible?

Line 407: please describe, why 10 epochs was used to train the model? Was it based on the performance (loss check)?

Line 420: Similar as above. The authors report that supervised classification training was performed for only one epoch with a learning rate of 1e-4 and batch size of 64. While early stopping was employed, it is unclear whether training for a single epoch is sufficient for the network to converge, particularly given that the classifier contains ~67k parameters. Typically, training deep models for such tasks requires multiple epochs to allow the optimization process to properly minimize the loss function and avoid underfitting. It would be important to clarify whether (i) additional epochs were tested, (ii) the training/validation curves indicated convergence within a single epoch, and (iii) performance stabilized consistently across runs. Otherwise, there is a risk that the reported accuracy reflects insufficient training rather than the true capacity of the model.

**Have the authors made all data and (if applicable) computational code underlying the findings in their manuscript fully available?**

Reviewer #1: Yes

Reviewer #2: Yes

Reviewer #3: **No: **

Reviewer #4: Yes

PLOS authors have the option to publish the peer review history of their article (what does this mean?). If published, this will include your full peer review and any attached files.

Reviewer #1: No

Reviewer #2: No

Reviewer #3: No

Reviewer #4: No

**Figure resubmission:**
---

## [Decision Letter · Decision Letter 1]

1 Dec 2025

Dear Dr. Walter,

We are pleased to inform you that your manuscript 'A Deep Learning approach for time-consistent cell cycle phase prediction from microscopy data' has been provisionally accepted for publication in PLOS Computational Biology.

Best regards,

Virginie Uhlmann

Academic Editor

PLOS Computational Biology

Stacey Finley

Section Editor

PLOS Computational Biology

Reviewer's Responses to Questions

**Comments to the Authors:**

Reviewer #1: Thanks for addressing all my comments

Reviewer #2: My comments have been sufficiently addressed with a extra experiments. I now support this manuscript's acceptance.

Reviewer #3: Manuscript Number: PCOMPBIOL-D-25-01257

Title: A Deep Learning approach for time-consistent cell cycle phase prediction from microscopy data

All my comments have been fully addressed in the revised manuscript. The authors added new experiments, clarified methodological details, expanded the introduction, improved usability, and updated figure quality. I recommend acceptance.

Reviewer #4: The authors addressed all the concerns and improved the manuscript.

**Have the authors made all data and (if applicable) computational code underlying the findings in their manuscript fully available?**

Reviewer #1: Yes

Reviewer #2: None

Reviewer #3: **No: **

Reviewer #4: Yes

PLOS authors have the option to publish the peer review history of their article (what does this mean?). If published, this will include your full peer review and any attached files.

Reviewer #1: No

Reviewer #2: No

Reviewer #3: No

Reviewer #4: No

---

## [Editor Report · Acceptance letter]

PCOMPBIOL-D-25-01257R1

A Deep Learning approach for time-consistent cell cycle phase prediction from microscopy data

Dear Dr Walter,

I am pleased to inform you that your manuscript has been formally accepted for publication in PLOS Computational Biology. Your manuscript is now with our production department and you will be notified of the publication date in due course.

With kind regards,

Anita Estes
